# Neural Persistence Dynamics

**Sebastian Zeng**[†/‡]**, Florian Graf**[†]**, Martin Uray**[†/‡]**, Stefan Huber**[‡]**, Roland Kwitt**[†]

[†]University of Salzburg, Austria
[‡]Josef Ressel Centre for Intelligent and Secure Industrial Automation,
University of Applied Sciences, Salzburg, Austria
{sebastian.zeng, florian.graf, roland.kwitt}@plus.ac.at,
{martin.uray, stefan.huber}@fh-salzburg.ac.at

## Abstract

We consider the problem of learning the dynamics in the topology of time-evolving point clouds, the prevalent spatiotemporal model for systems exhibiting collective behavior, such as swarms of insects and birds or particles in physics. In such systems, patterns emerge from (local) interactions among self-propelled entities. While several well-understood governing equations for motion and interaction exist, they are notoriously difficult to fit to data, as most prior work requires knowledge about individual motion trajectories, i.e., a requirement that is challenging to satisfy with an increasing number of entities. To evade such confounding factors, we investigate collective behavior from a *topological perspective*, but instead of summarizing entire observation sequences (as done previously), we propose learning a latent dynamical model from topological features *per time point*. The latter is then used to formulate a downstream regression task to predict the parametrization of some a priori specified governing equation. We implement this idea based on a latent ODE learned from vectorized (static) persistence diagrams and show that a combination of recent stability results for persistent homology justifies this modeling choice. Various (ablation) experiments not only demonstrate the relevance of each model component but provide compelling empirical evidence that our proposed model – *Neural Persistence Dynamics* – substantially outperforms the state-of-the-art across a diverse set of parameter regression tasks.

## 1 Introduction

Understanding emerging behavioral patterns of a collective through the interaction of individual entities is key to elucidate many phenomena in nature on a macroscopic and microscopic scale. Prominent examples are coherently moving flocks of birds, the swarming behavior of insects and fish or the development of cancerous cells, all understood as 2D/3D point clouds that *evolve over time*. Importantly, several widely-accepted governing equations for collective behavior exist [18, 40, 56] which, *when appropriately parameterized*, can reproduce different incarnations of typically observed patterns. Even more importantly, these equations are tied to physically interpretable parameters and can provide detailed insights into the intrinsic mechanisms that control various behavioral regimes. However, while it is fairly straightforward to simulate collective behavior from governing equations (see [21]), the *inverse* problem, i.e., *identifying* the model parameters from the data, turns out to be inherently difficult. Confounding factors include the often large number of observed entities and the difficulty of reliably identifying individual trajectories across point clouds at possibly non-equidistant observation times.

However, as several works [4, 23, 52] have recently demonstrated, it may not be necessary to rely on individual trajectories for parameter identification. In fact, collective behavior is characterized by global patterns that emerge from local interactions, and we observe the emergence of these patterns through changes to the "shape" of point clouds over time. For instance, birds may form a flock, split

38th Conference on Neural Information Processing Systems (NeurIPS 2024).

into groups, and then merge again. This perspective has prompted the idea of summarizing such topological events over time and then phrasing model identification as a downstream *prediction/regression task*. The key challenge here lies in the transition from topological summaries of point clouds at specific observation times, typically obtained via *persistent homology (PH)* [3, 9, 22], to the *dynamic* regime where the temporal dimension plays a crucial role.

Yet, despite the often remarkable performance of topological approaches in terms of parameter identification for models of collective behavior, it remains unclear in which situations they are preferable over more traditional (learning) methods that can handle point cloud data, as, e.g., used in computer vision problems [24, 46]. In the spirit of [53], we highlight this point by previewing a snapshot of one ablation experiment from Sec. 4. In particular, Tbl. 1 compares the parameter regression performance of our proposed approach, using PH, to a variant

**Table 1:** PointNet++ [45] vs. persistent homology (PH) representations; ↑ means higher and ↓ means lower is better.

| | ⊘ **VE** ↑ | ⊘ **SMAPE** ↓ |
|---|---|---|
| | vicsek-10k | |
| Ours (PointNet++, v2) | 0.274±0.085 | 0.199±0.014 |
| Ours (PH, v1) | **0.579**±0.034 | **0.146**±0.006 |
| | dorsogna-1k | |
| Ours (PointNet++, v2) | 0.816±0.031 | 0.132±0.018 |
| Ours (PH, v1) | **0.851**±0.008 | **0.096**±0.004 |

where we instead use PointNet++ [45] representations. Referring to Fig. 1, this means that the $\mathbf{v}_{\tau_i}$ are computed by a PointNet++ model. As can be seen in Tbl. 1, relying on representations computed via PH works well on *both datasets*, while PointNet++ representations fail on one dataset (vicsek-10k), but indeed perform reasonably well on the other (dorsogna-1k).

In light of this observation, it is worth emphasizing that there is a clear distinction in terms of the *source* of point cloud dynamics when comparing collective behavior to problems in computer vision where moving objects are of primary interest and PointNet variants (see [24]) have shown stellar performance: in particular, point clouds in vision primarily evolve due to a change in pose or camera motion, whereas collective behavior is driven by (local) intrinsic point interactions. The latter induces changes to the "shape" of the point clouds, i.e., a phenomenon that is typically not observed in vision.

**Contribution(s).** Our key idea is to learn a generative model that can reproduce topological summaries *at each point in time*. This contrasts prior work, which primarily aims to extract *one* summary representation of the entire timeline. In detail, we advocate for modeling the dynamics of vectorized persistence diagrams via a continuous latent variable model (e.g., a latent ODE), see Fig. 1, and to use the resulting latent paths as input to a downstream parameter regression task. Recent stability results for vectorizations of persistence diagrams – relating distances among the latter to the Wasserstein distance between point clouds – justify this modeling choice. Aside from state-of-the-art performance on various parameter identification tasks for established models of collective behavior, our approach scales favorably with the number of observed sequences, accounts for non-equidistant observation times, and is easily combinable with other sources of information.

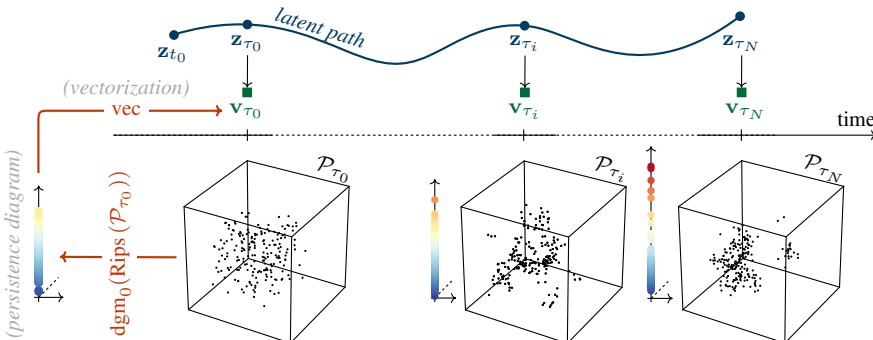

**Figure 1:** Conceptual overview of *Neural Persistence Dynamics*. Given is a sequence of observed point clouds $\mathcal{P}_{\tau_0}, \ldots, \mathcal{P}_{\tau_N}$. *First*, we summarize each $\mathcal{P}_{\tau_i}$ via (Vietoris-Rips) persistent homology into persistence diagrams (zero-, one- and two-dimensional; only the zero-dimensional diagrams are shown) which are then vectorized into $\mathbf{v}_{\tau_i}$ via existing techniques. *Second*, we model the dynamics in the sequence $\mathbf{v}_{\tau_0}, \ldots, \mathbf{v}_{\tau_N}$ via a continuous latent variable model (in our case, a latent ODE) and then use a summary of the latent path to predict the *parameters* of specific governing equation(s) of collective behavior. Precomputed steps are highlighted in red.

## 2   Related work

Our work is partially related to the literature on learning with dynamic point clouds in vision problems, but *primarily* connects to work on summarizing topological features over time and inferring interaction laws of collective behavior from data.

**Summarizing topological features over time.** A common denominator in the literature on encoding topological changes in dynamic point clouds is the use of *persistent homology*, extended to accommodate the temporal dimension in various ways. One of the early works along this direction are *persistence vineyards* [16], introduced as a means to study folding trajectories of proteins. Based on the idea of tracking points in persistence diagrams over time, vineyards come with stability properties similar to persistent homology [41], but are expensive to compute and compare on a large scale. In cases where one would know the "right" scale at which to compute homology, one may also use *zigzag persistence* [8], as done in [17, 54], to track homological changes over time. Nevertheless, for problems with a large number of observation sequences, scale selection per sequence is nontrivial and highly impractical.

Alternatively, instead of thinking about the evolution of individual points in persistence diagrams over time, one may discard the matching between points and instead focus on *sequences of summary representations* of said diagrams. In [25], for instance, the authors work directly with persistence diagrams (per time point) to identify changes in the topology of time-varying graphs. In terms of temporal summary representations, [52] introduce *crocker plots* to encode the evolution of topological features by stacking discretized Betti curves over time. *Crocker stacks* [57], an extension of this concept, adds a smoothing step that gradually reduces the impact of points of low persistence and, upon discretization, yields a third dimension to crocker plots. In our context, both crocker plots & stacks have been successfully used as input to regression methods to estimate the parametrization of models of collective behavior [4]. By drawing on prior work on kernels for sequentially ordered data [33], [23] follow a conceptually similar strategy as [52, 57], introducing a *path signature kernel (PSK)* for sequences of summary representations of persistence diagrams.

Along a different line of research, [28, 29] propose *formigrams* as summaries of dynamic metric data, encoding the evolution of *connected components*. In subsequent work, [30] construct multidimensional (i.e., spatiotemporal) persistent homology modules from dynamic metric data and compare invariants of these modules. While these works provide important theoretical stability results for zero-dimensional homological features, i.e., connected components, it is unclear how their construction extends to homological features of higher dimension in a tractable manner. However, it is worth pointing out that recent progress along the lines of vectorizations of multiparameter persistent homology [36] in combination with [30] (who essentially construct multiparameter persistence modules) might, in the future, constitute a tractable approach to study dynamically changing point clouds with learning methods.

Notably, computational challenges also arise in the context of crocker plots/stacks and PSKs. Despite their remarkable performance in distinguishing different configurations of models for collective behavior, both approaches suffer scalability issues: either (i) in terms of unfavorable scalability with respect to the dimensionality of vectorized persistence diagrams (as with the PSK approach of [23]), or (ii) in terms of unfavorable scalability with the number of observation sequences (as is the case for crocker plots/stacks, due to the need for extensive cross-validation of the discretization parameters). As we show in Sec. 4, our method not only outperforms these techniques by a large margin, but also scales to large amounts of training sequences and requires little hyperparameter tuning.

In addition to the closely related works discussed above, we highlight that there is a large body of literature on the topological analysis of time-varying signals, such as studying fMRI data via cubical persistence [47], or the persistent homology of delay embeddings [43, 44] of time series. In our context, however, these works are only partially related/relevant, as they do assume precise knowledge about individual trajectories over time (e.g., voxel IDs in fMRI data), which is unrealistic when seeking to infer parametrizations of models for collective behavior from data.

**Inferring interaction laws for models of collective behavior.** In the context of studying characteristics of collective behavior, there is a second line of closely related work on inferring interaction laws (see, e.g., [2, 5, 6, 27, 39, 61]), ranging from metric-distance-based models and topological interaction models to non-parametric estimators of interaction kernels. While a thorough survey of this literature is beyond the scope of this paper, we highlight that one *common denominator* in

these works is their reliance on correspondences between points across time, e.g., to infer individual point velocities or trajectories. To give an example, in recent work [39, 61], the authors derive non-parametric estimators for interaction kernels (certain functions of pairwise distances) between observed points which crucially hinges on the traceability of each particle over time. While such approaches are conceptually appealing and even allow for reconstructing or extrapolating trajectories, they operate in the *observation space*. Even under a moderate number of points $m$, computing these estimators becomes prohibitively expensive (as, e.g., in 3D, the state space is $\mathbb{R}^{3m}$). In contrast, our approach only requires positional information and can handle larger point clouds. Furthermore, although we only present results on predicting parameters for a class of a priori specified governing equations, by formulating an auxiliary regression task, our underlying model may also be used to predict other quantities of interest.

Finally, taking a slightly broader perspective on prior art, we want to highlight recent progress on learning-based approaches that study inverse problems in the context of classic partial differential equations (PDEs). It might be possible to apply such approaches [37, 55, 60] to specific models of collective behavior such as volume exclusion [40], for which the asymptotics of infinitely many particles are solutions to parametric PDEs [42]. However, for other models, the number of particles strongly influences the dynamics. For instance, in the D'Orsogna model [18], particle distances can collapse to zero as the number of particles tends to infinity.

## 3 Method

Below, we present our method with a focus on its application to modeling collective behavior in point clouds. Importantly, the latter represents only one of many conceivable use cases. In fact, the core ideas can be applied in exactly the same way whenever one can extract topological features from time-dependent data (e.g., graphs or images).

**Notation**. In the following, $\mathcal{P} = \{\mathbf{x}_1, \ldots, \mathbf{x}_M\} \subset \mathbb{R}^3$ denotes a point cloud with $M$ points and $d(\mathbf{x}, \mathbf{y}) = \|\mathbf{x} - \mathbf{y}\|$ denotes the Euclidean metric. Point clouds may be indexed by $t_i$ (or $\tau_i$) to highlight the dependence on time $t_i$. If necessary, we clearly distinguish between $\tau_i$ as a time point with an available observation, and $t_i$ as a general placeholder for time.

**Problem statement**. Given a sequence of point clouds $\mathcal{P}_{\tau_0}, \ldots, \mathcal{P}_{\tau_N}$, observed at possibly non-equidistant time points $\tau_i$, we (1) seek to model their topological evolution over time and then (2) use this model to predict the parametrization of an a priori defined governing equation of collective behavior. The latter is typically specified by a small number of parameters $\beta_1, \ldots, \beta_P$ that control the motions $d\mathbf{x}_i/dt$ of individual points $\mathbf{x}_i$ and specify (local) interactions among neighboring points.

As preparation for our model description, we first briefly establish how one may extract topological features from a point cloud $\mathcal{P}$, using *persistent homology* [3, 9, 22] – the arguably most prominent and computationally most feasible approach.

**Persistent homology of point clouds.** Persistent homology seeks to uncover and concisely summarize topological features of $\mathcal{P}$. To this end, one constructs a topological space from $\mathcal{P}$ in the form of a simplicial complex, and studies its homology across multiple scales. The most relevant construction for our purposes is the *Vietoris-Rips* complex $\mathrm{Rips}(\mathcal{P})_\delta$, with vertex set $\mathcal{P}$. This complex includes an $m$-simplex $[\mathbf{x}_0, \ldots, \mathbf{x}_m]$ iff $d(\mathbf{x}_i, \mathbf{x}_j) \leq \delta$ for all $0 \leq i, j \leq m$ at a given threshold $\delta$. The "shape" of this complex can then be studied using homology, a tool from algebraic topology, with zero-dimensional homology ($\mathrm{H}_0$) encoding information about connected components, one-dimensional homology ($\mathrm{H}_1$) encoding information about loops and two-dimensional homology ($\mathrm{H}_2$) encoding information about voids; we refer to this information as *homological/topological features*. Importantly, if $\delta_b \leq \delta_d$, then $\mathrm{Rips}(\mathcal{P})_{\delta_b} \subseteq \mathrm{Rips}(\mathcal{P})_{\delta_d}$, inducing a sequence of inclusions when varying $\delta$, called a *filtration*. The latter in turn induces a sequence of vector spaces $\mathrm{H}_k(\mathrm{Rips}(\mathcal{P})_{\delta_b}) \to \mathrm{H}_k(\mathrm{Rips}(\mathcal{P})_{\delta_d})$ at the homology level. Throughout this sequence, homological features (corresponding to basis vectors of the different vector spaces) may appear and disappear; we say they are *born* at some $\delta_b$ and *die* at $\delta_d > \delta_b$. For instance, a one-dimensional hole might appear at a particular value of $\delta_b$ and disappear at a later $\delta_d$; in other words, the hole *persists* from $\delta_b$ to $\delta_d$, hence the name *persistent homology*. As different features may be born and die at the same time, the collection of (birth, death) tuples is a multiset of points, often represented in the form of a persistence diagram, which we denote as $\mathrm{dgm}_k(\mathrm{Rips}(\mathcal{P}))$. Here, the notation $\mathrm{Rips}(\mathcal{P})$ refers to the full filtration and $\mathrm{dgm}_k$ indicates that we have tracked $k$-dimensional homological features throughout this filtration.

Clearly this construction does not account for any temporal changes to a point cloud, but reveals features that are present at a *specific* time point. Furthermore, due to the inconvenient multiset structure of persistence diagrams for learning problems, one typically resorts to appropriate *vectorizations* (see, e.g., [1, 7, 10, 26]), all accounting for the fact that points close to the diagonal $\Delta = \{(x,x) : x \in \mathbb{R}\}$ contribute less (0 at $\Delta$) to the vectorization. The latter is important to preserve stability (see paragraph below) with respect to perturbations of points in the diagram. In the following, we refer to a vectorized persistence diagram of a point cloud $\mathcal{P}_{t_i}$ as

$$\mathbf{v}_{t_i,k} := \text{vec}(\text{dgm}_k(\text{Rips}(\mathcal{P}_{t_i}))) \ , \tag{1}$$

where $\mathbf{v}_{t_i,k} \in \mathbb{R}^d$ and $d$ is controlled by hyperparameter(s) of the chosen vectorization technique. For brevity, we omit the subscript $k$ when referring to vectorizations unless necessary.

**Remark 1.** *As our goal is to model the dynamics of vectorized persistence diagrams using a continuous latent variable model, it is important to discuss the dependence of the vectorizations on the input data. In particular, for our modeling choice to be sound, vectorizations should vary (Lipschitz) continuously with changes in the point clouds over time. We discuss this aspect next.*

**Stability/Continuity aspects.** First, we point out that persistence diagrams can be equipped with different notions of *distance*, most prominently the bottleneck distance and Wasserstein distances, both based upon the cost of an optimal matching between two diagrams, allowing for matches to the diagonal $\Delta$. We refer the reader to [12, 51] for a detailed review. *Stability* in the context of persistent homology is understood as persistence diagrams varying (Lipschitz) continuously with the input data. While seminal stability results exist for the Wasserstein distances [15] and the bottleneck distance [12, 14], most results from the literature focus on the latter.

In the case of *vectorization techniques* for persistence diagrams (e.g., [1, 26]), it turns out that, most of the time, vectorizations are only Lipschitz continuous with respect to Wasserstein distances, and existing results are typically of the form

$$d(\text{vec}(F), \text{vec}(G)) \leq K\, W_1(F, G) \ , \tag{2}$$

where $W_1$ denotes the $(1, 2)$-Wasserstein distance as defined in [51, Def. 2.7], $F, G$ are two persistence diagrams and $K > 0$ is a Lipschitz constant. In the context of Rem. 1, one may be tempted to combine Eq. (2) with the seminal *Wasserstein stability theorem* from [15] to infer stability of vectorizations with respect to the input data. Yet, the conditions imposed in [15] actually eliminate a direct application of this result in most practical cases, as discussed in [51]. However, the latter work also provides an alternative: in our particular case of Vietoris-Rips persistent homology, one can upper bound $W_1$ in terms of the standard *point set* Wasserstein distance $\mathcal{W}_1$. Specifically, upon relying on a stable vectorization technique, we obtain the inequality chain

$$d(\text{vec}(F_k), \text{vec}(G_k)) \overset{[26,\ \text{Thm. 12}]}{\leq} K\, W_1(F_k, G_k) \\ \overset{[51,\ \text{Thm. 5.9}]}{\leq} 2K \binom{M-1}{k} \mathcal{W}_1(\mathcal{P}, \mathcal{Q}) \ , \tag{3}$$

where $F_k = \text{dgm}_k(\text{Rips}(\mathcal{P}))$, $G_k = \text{dgm}_k(\text{Rips}(\mathcal{Q}))$ denote the $k$-dimensional persistence diagrams of point clouds $\mathcal{P}$ and $\mathcal{Q}$ of equal cardinality $M$. In particular, we realize vec via the approach outlined in [26, Def. 10] using *exponential structure elements* [26, Def. 19].

Overall, the continuity property in Eq. (3) guarantees that small changes in dynamic point clouds over time only induce small changes in their vectorized persistence diagrams, and therefore provides a solid justification for the model discussed next.

**Latent variable model for persistence diagram vectorizations.** As we presume that the dynamic point clouds under consideration are produced (or can be described sufficiently well) by equations of motions with only a few parameters, cf. Fig. 3, it is reasonable to assume that the dynamics of the (vectorized) persistence diagrams are equally governed by a somewhat simpler unobserved/latent dynamic process in $\mathbb{R}^z$ with $z \ll d$. We model this latent dynamic process via a neural ODE [13, 49], learned in a variational Bayes regime[1] [32]. In this setting, one chooses a recognition/encoder network ($\text{Enc}_{\boldsymbol{\theta}}$) to parametrize an approximate variational posterior $q_{\boldsymbol{\theta}}(\mathbf{z}_{t_0}|\{\mathbf{v}_{\tau_i}\}_i)$, an ODE solver

---

[1]Other choices, e.g., a latent SDE as in [35, 58] are possible, but we did not explore this here.

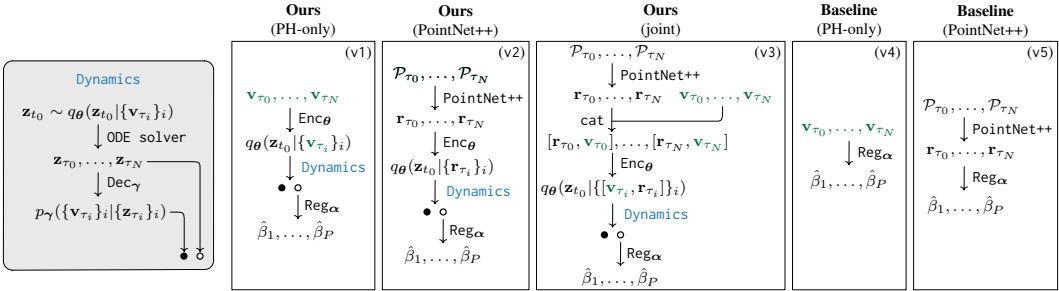

**Figure 2:** Schematic illustration of different model variants. The first three variants (*left* to *right*) explicitly model latent dynamics (later denoted as *w/ dynamics*), the baseline variants do not (later denoted as *w/o dynamics*), but still incorporate the attention mechanism of the encoder from [50], which we use throughout.

to yield latent states $\{\mathbf{z}_{\tau_i}\}_i$ at observed time points $\tau_i \in [0, T]$ and a suitable generative/decoder network ($\mathsf{Dec}_{\gamma}$) to implement the likelihood $p_{\gamma}(\mathbf{v}_{\tau_i}|\mathbf{z}_{\tau_i})$. Upon choosing a suitable prior $p(\mathbf{z}_{t_0})$, one can then train the model via ELBO maximization, i.e.,

$$\boldsymbol{\theta}, \boldsymbol{\gamma} = \arg\max_{\boldsymbol{\theta},\boldsymbol{\gamma}} \mathbb{E}_{\mathbf{z}_{t_0}\sim q_{\boldsymbol{\theta}}} \Big[ \sum_i \log p_{\boldsymbol{\gamma}}(\mathbf{v}_{\tau_i}|\mathbf{z}_{\tau_i}) \Big] - D_{\mathrm{KL}}(q_{\boldsymbol{\theta}}(\mathbf{z}_{t_0}|\{\mathbf{v}_{\tau_i}\}_i) \| p(\mathbf{z}_{t_0})) \ . \tag{4}$$

Different to [49], we do not implement the recognition/encoder network via another neural ODE, but rather choose an attention-based approach (mTAN) [50] which can even be used in a standalone manner as a strong baseline (see Sec. 4). In our implementation, the recognition network yields the parametrization $(\boldsymbol{\mu}, \boldsymbol{\Sigma})$ of a multivariate Gaussian in $\mathbb{R}^z$ with diagonal covariance, and the prior is a standard Gaussian $\mathcal{N}(\mathbf{0}, \mathbf{I}_z)$. Furthermore, the ODE solver (e.g., Euler) can yield $\mathbf{z}_{t_i}$ at any desired $t_i$, however, we can only evaluate the ELBO at observed time points $\tau_i$.

**Regression objective**. To realize our downstream regression task, i.e., predicting parameters $\beta_1, \ldots, \beta_P$ of an underlying governing equation for collective behavior (see Fig. 3) from a given observation sequence, we have multiple choices. By our assumption of a latent dynamic process that carries information about the dynamic nature of the topological changes, it is reasonable to tie the simulation parameter estimates $\hat{\beta}_1, \ldots, \hat{\beta}_P$ to the *latent path* $\{\mathbf{z}_{t_i}\}_i$ via a regression network $\mathsf{Reg}_{\boldsymbol{\alpha}}$ that accepts $\{\mathbf{z}_{t_i}\}_i$ as input. In particular, we re-use the attention-based encoder architecture $\mathsf{Enc}_{\boldsymbol{\theta}}$ to allow attending to different parts of this path. However, different to $\mathsf{Enc}_{\boldsymbol{\theta}}$, which parametrizes the approximate posterior from *observations* at time points $\tau_i$, the regression network $\mathsf{Reg}_{\boldsymbol{\alpha}}$ (with its own set of parameters $\boldsymbol{\alpha}$) receives *latent states* $\mathbf{z}_{t_i}$ at equidistant $t_i \in [0, T]$, then summarizes this sequence into a vector and linearly maps the latter to $\hat{\beta}_1, \ldots, \hat{\beta}_P$, cf. Fig. 2. As in [49], for training, we extend the ELBO objective of Eq. (4) by an additive auxiliary regression loss, such as the mean-squared error (MSE), between the predictions $\hat{\beta}_p$ and the ground truth $\beta_p$, implicitly making a Gaussian noise assumption.

A schematic overview of our Neural Persistence Dynamics approach is shown in Fig. 1. Additional details, including different model variants, are illustrated in Fig. 2.

**Remark 2.** *Clearly, our presented framework allows for many architectural choices. While some components affect downstream (regression) performance only marginally, others have a more profound impact, and we have already identified some recommended choices above (based on our experiments in Sec. 4). This includes (1) our choice of a stable persistent homology vectorization* vec, *where we rely on [26] due to consistently reliable performance without much hyperparameter tuning, and (2) our choice of recognition network, where we choose an attention-based approach (mTAN) [50] which has proven to be very effective in practice [58]. In Sec. 4.2, we will provide additional configuration details for our experimental study.*

## 4 Experiments

### 4.1 Datasets

Similar to previous work [4, 23, 52, 57], we evaluate and compare our approach on simulation data that is generated from parametric models of collective behavior. Specifically, we consider

**D'Orsogna** (dorsogna) [18]

$$\frac{\mathrm{d}\mathbf{x}_i}{\mathrm{d}t} = \mathbf{v}_i, \; m\frac{\mathrm{d}\mathbf{v}_i}{\mathrm{d}t} = \left(\alpha - \beta |\mathbf{v}_i|^2\right)\mathbf{v}_i - \frac{1}{M}\nabla_{\mathbf{x}_i}\sum_{j=1,j\neq i}^{M} U(\|\mathbf{x}_i - \mathbf{x}_j\|), \; U(r) = C_r e^{-\frac{r}{l_r}} - C_a e^{-\frac{r}{l_a}} \quad (5)$$

**Vicsek** (vicsek) [20, 56]

$$\frac{\mathrm{d}\mathbf{x}_i}{\mathrm{d}t} = c\,\mathbf{v}_i, \; \mathrm{d}\mathbf{v}_i = \left(\mathbf{I} - \mathbf{v}_i\mathbf{v}_i^\top\right)\left(\nu\,\bar{\mathbf{v}}_i\,\mathrm{d}t + \sqrt{2D}\,\mathrm{d}\mathbf{B}_t\right), \; \bar{\mathbf{v}}_i = \frac{\mathbf{a}_i}{\|\mathbf{a}_i\|}, \; \mathbf{a}_i = \sum_{j,\|\mathbf{x}_i - \mathbf{x}_j\|\leq R}\mathbf{v}_j \quad (6)$$

**Volume exclusion** [40] (volex) variant from [21]

$$\frac{\mathrm{d}\mathbf{x}_i}{\mathrm{d}t} = -\frac{\alpha}{R}\sum_{j=1,i\neq j}^{M}\phi\left(\frac{\|\mathbf{x}_j - \mathbf{x}_i\|^2}{4R^2}\right)(\mathbf{x}_i - \mathbf{x}_j), \; \text{with} \; \phi(r) = \begin{cases} \frac{1}{r} - 1, & 0 < r \leq 1 \\ 0, & \text{else} \end{cases} \quad (7)$$

$+$ *cell division* & *cell death* at constant rate(s) $\lambda_b, \lambda_d \in [0,1]$

**Figure 3:** Models of collective behavior. Parameters that are varied to obtain different behavior are highlighted in red; the range of each parameter is listed in Appendix A. In the **Vicsek** model, $\mathbf{B}_t$ denotes Brownian motion.

three different models in $\mathbb{R}^3$: D'Orsogna [18], Vicsek [20] and volume exclusion [40], using the publicly-available implementations in the SiSyPHE library [21][2]. The corresponding equations of motion (and interaction laws) are summarized in Fig. 3. The parameters that are varied to generate the datasets, i.e., the response variables of the regression tasks, are highlighted in red. We sample these parameters, as specified in Appendix A, to cover a wide range of macroscopic behavioral regimes. For each sampled parameter configuration, we simulate one sequence of point clouds, to a total of 10,000 sequences per model. All simulations are run for 1,000 time steps (with step size 0.01, starting at $t = 0$) on point clouds of size $M = 200$. We take every 10th time step as an observation, yielding observation sequences of length 100. At $t = 0$, points are drawn independently and uniformly in $[-0.5, 0.5]^3$, and initial velocities are uniformly distributed on the unit sphere. For direct comparison to [23], we also simulated their parameter configuration of the D'Orsogna model. This setup yields four datasets: dorsogna-1k (from [23]), dorsogna-10k, vicsek-10k and volex-10k.

### 4.2 Implementation, training & evaluation metrics

**Implementation.** The model variants from Fig. 2 can be realized in many ways. Below, we specify the configuration that was used to run the experiments. For the encoder $\mathsf{Enc}_{\boldsymbol{\theta}}$, as well as the regression network $\mathsf{Reg}_{\boldsymbol{\alpha}}$, we use the attention-based mTAN architecture [50]. As decoder network $\mathsf{Dec}_{\boldsymbol{\gamma}}$, we choose a two-layer MLP with ReLU activations, and as ODE solver, we select the Euler method. Each model is trained for 150 epochs using ADAM [31] (with a weight decay of 0.001), starting at a learning rate of 0.001 (decaying according to a cosine annealing schedule) and MSE as a reconstruction (i.e., to evaluate the first term in Eq. (4)) and regression loss. In case a model uses topological features, we use Ripser++ [59] to compute (prior to training) persistent homology of dimension up to two, i.e., $H_0$, $H_1$, and $H_2$. Vectorizations of *each* persistence diagram are then obtained using *exponential structure elements* from [26, Def. 19]. In particular, we use 20 structure elements per diagram, which yields a $d = 3 \cdot 20$ dimensional representation per point cloud and time point. The location of each structure element is set to one of 20 $k$-means++ cluster centers, obtained by running the latter on a random subset of 50,000 points (per dimension) selected from all persistence diagrams available during training; the scale parameter of each structure element is set according to [48, Eq. (2)]. To model the dynamics, we fix the latent space dimensionality to $z = 20$ and scale the ODE integration time to $[0, 1]$. While other settings are undoubtedly possible, we did not observe any noticeable benefits from increasing the dimensionality of the vectorizations or the latent space. Our publicly available reference implementation can be found at https://github.com/plus-rkwitt/neural_persistence_dynamics.

**Evaluation metrics.** We randomly partition each dataset into five training/testing splits of size 80/20. To obtain a robust performance estimate for different regimes of missing and unevenly spaced observations, we train three models (per split) using only a fraction (i.e., 20%, 50%, and 80%) of

---

[2]https://github.com/antoinediez/Sisyphe

randomly chosen time points per sequence. Similarly, all testing splits undergo the same sampling procedure. All scores (see below) are reported as an average ($\pm$ one standard deviation) over the five splits and the three time point sampling percentages. Specifically, we report the *variance explained (VE)* [34] (in $[0, 1]$; higher is better $\uparrow$) and the *symmetric mean absolute percentage error (SMAPE)* (in $[0, 1]$; lower is better $\downarrow$). For each testing sequence in each split, the scores are computed from the *true* simulation parameters (see variables marked red in Fig. 3) and the corresponding *predictions* (denoted by $\hat{\beta}_i$ in Fig. 2). Finally, when reporting results on a dataset, we mark the best score in **bold**, as well as all other scores that *do not* show a statistically significant difference in mean (assessed via a Mann-Whitney test at 5% significance and correcting for multiple comparisons).

## 4.3 Ablation

In our ablation study, we assess (1) the relevance of different point cloud representations (PH vs. PointNet++), (2) any potential benefits of modeling latent dynamics and (3) the impact of varying the observation timeframe. Additional ablation experiments can be found in Appendix B.3.

**Are representations complementary?** We first investigate the impact of different point cloud representations (from PH and PointNet++, resp.), by comparing variants `v1`,`v2` and `v3` from Fig. 2.

Tbl. 2 shows an extension of Tbl. 1, listing results for the `vicsek-10k` and `dorsogna-1k` data. While using PointNet++ representations on `vicsek-10k` yields rather poor performance, combining them with representations computed from PH is at least *not* detrimental. On the other hand, on `dorsogna-1k`, where PointNet++ yields decent SMAPE and VE scores, the combination of both sources substantially outperforms each single source in isolation. Although, the complementary nature of a topological perspective on data has been pointed out many times in the literature, it is rarely as pronounced as in this

**Table 2:** Ablation study on the relevance of different point cloud representations.

| Source | ⊘ **VE** ↑ | ⊘ **SMAPE** ↓ |
|---|---|---|
| | `vicsek-10k` | |
| PointNet++ (v2) | 0.274±0.085 | 0.199±0.014 |
| PH (v1) | **0.579**±0.034 | **0.146**±0.006 |
| PH+PointNet++ (v3) | **0.576**±0.030 | **0.144**±0.006 |
| | `dorsogna-1k` | |
| PointNet++ (v2) | 0.816±0.031 | 0.132±0.018 |
| PH (v1) | 0.851±0.008 | 0.097±0.005 |
| PH+PointNet++ (v3) | **0.931**±0.004 | **0.067**±0.002 |

particular experiment. *Hence, we will stick to the combination of PH+PointNet++ representations (i.e.,* `v3` *in Fig. 2) in any subsequent experiments.*

**Are explicit latent dynamics beneficial?** To assess the impact of explicitly modeling the dynamics of persistence diagram vectorizations, we ablate the latent ODE part of our model. In detail, we compare `v3` from Fig. 2 against using $\text{Reg}_\alpha$ operating *directly* on concatenated point cloud representations from PH and PointNet++ (i.e., a combination of variants `v4` & `v5` from Fig. 2). The latter approach constitutes an already *strong baseline* (cf. [58]) as $\text{Reg}_\alpha$ incorporates an attention mechanism that allows attending to relevant parts of each sequence. For a fair comparison, we increase the size of $\text{Reg}_\alpha$ to approximately match the number of parameters to our latent dynamic model.

As in Tbl. 2, we list results for the `vicsek-10k` and `dorsogna-1k` data in Tbl. 3. First, we see that explicitly modeling the latent dynamics is beneficial, with considerable improvements in the reported scores across both datasets. While differences are quite prominent on the `dorsogna-1k` data, they are less pronounced in the SMAPE score on `vicsek-10k`, but still statistically significant. Second, it is important to point out that even without any explicit latent dynamics, the regression performance (see Tbl. 4) is already above the current state-of-the-art

**Table 3:** Results on the relevance of latent dynamics.

| | ⊘ **VE** ↑ | ⊘ **SMAPE** ↓ |
|---|---|---|
| | `vicsek-10k` | |
| w/ dynamics | **0.576**±0.030 | **0.144**±0.006 |
| w/o dynamics | 0.512±0.020 | 0.155±0.004 |
| | `dorsogna-1k` | |
| w/ dynamics | **0.931**±0.004 | **0.067**±0.002 |
| w/o dynamics | 0.850±0.006 | 0.098±0.004 |

(i.e., compared to PSK [23]), highlighting the necessity for strong baselines.

**Impact of the observation timeframe.** So far, we have presented results (as in prior work) where observation sequences are extracted from the beginning of a simulation (i.e., starting from $t_0 = 0$), which corresponds to observing the emergence of patterns out of a random yet qualitatively similar, initial configuration of points. In the following experiment, we investigate the impact of extracting observation sequences with *starting points randomly spread across a much longer simulation time*.

This is relevant as it is unlikely to observe uniform initial positions in any data obtained from real-world experiments. To this end, we create variations of the `dorsogna-10k` data by progressively increasing the simulation time $T$ from 1,000 to 20,000 and then randomly extracting sub-sequences of length 1,000 (again, as before, taking each 10th step as an observation). The extension of the simulation timeframe has the effect that the difficulty of the learning task considerably increases with $T$. We argue that this is due to the increased variation across the observed sequences while the amount of training data remains the same. For comparison, we also list results for the PSK approach of [23], however, only for the *optimal* case of observing *all* time points within a sequence (due to the massive PSK computation time; $> 3$ days). As the PSK relies on persistent homology only, we limit our model to the `v1` variant from Fig. 2. As seen from Fig. 4, we observe an increase/drop in SMAPE and VE, resp., for both approaches, yet our approach degrades much slower with $T$.

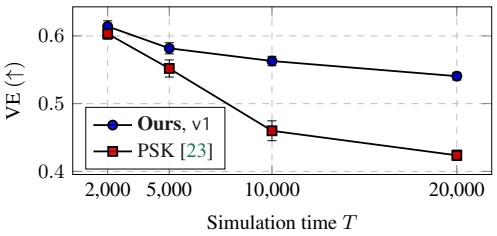 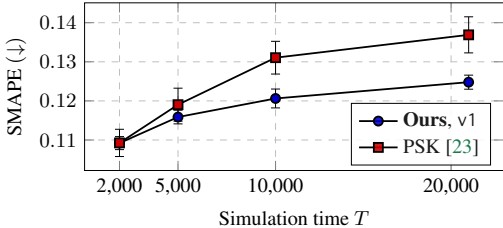

**Figure 4:** Impact of the maximal simulation time $T$ for extracting training/testing sequences starting at $\tau_0 \in [0, T - 1000]$, assessed on the `dorsogna-10k` dataset.

### 4.4 Comparison to the state-of-the-art

Finally, we present parameter regression results on the full battery of datasets and compare against two of the most common approaches from the literature: the *path signature kernel (PSK)* of [23] and the *crocker stacks* approach from [57]. For both competitors, we report results when *all* time points are observed to establish a baseline of *what could be achieved* in the best case. Note that we evaluate the PSK approach [23] on exactly the same vectorizations as our approach, and we replicate their experimental setting of choosing the best hyperparameters via cross-validation. Similarly, for crocker stacks, we cross-validate the discretization parameters, see Appendices B.1 and B.2. Tbl. 4 lists the corresponding results. Aside from the observation that our approach largely outperforms the state-of-the-art across all parameter regression tasks, we also highlight that our model is trained with *exactly* the same hyperparameters on all datasets.

**Remark 3.** *A closer look at the* `volex-10k` *dataset, in particular its governing equations in Fig. 3, shows that the cardinality of the point clouds may change over time due to cell division or death. While this is no practical limitation for our approach, our stability arguments from Eq. (3) no longer apply. We conjecture that one may be able to extend the result of [51] to account for such cases, but this might require a case-by-case analysis and possibly include additional assumptions.*

## 5 Discussion

We introduced a framework to capture the dynamics observed in topological summary representations over time and presented one incarnation using a latent ODE on top of vectorized Vietoris-Rips persistence diagrams, addressing the problem of *predicting parametrizations for models of collective behavior*. Conceptually, *Neural Persistence Dynamics* embodies the idea of learning the dynamics in a temporal sequence of *vectorized* topological summaries instead of, e.g., trying to track individual homology classes over time (as with persistence vineyards, see Sec. 2). Our approach successfully scales to a large number of observation sequences, requires little to no parameter tuning, and vastly outperforms the current state-of-the-art, as demonstrated by several experiments on dynamic point cloud datasets with varying characteristics. Finally, we emphasize that the fundamental ideas of *Neural Persistence Dynamics* may also be applied to other types of time-varying data (e.g., graphs or images) and downstream objectives simply by adjusting the filtration choice. We hope that our work will stimulate further research in this direction.

**Table 4:** Comparison to the state-of-the-art across four diverse datasets of collective behavior. Ours (joint) refers to the variant v3 of Fig. 2, Ours (PH-only) refers to variant v1. The latter setting is directly comparable to the PSK and crocker stacks approach. Best results are marked **bold**. Multiple bold entries indicate that there is no significant difference in mean. Note that `dorsogna-1k` is simulated exactly as in [23] varying only *two parameters*, whereas for `dorsogna-10k` we vary *four* parameters (as with `vicsek-10k` and `volex-10k`).

| | | $\oslash$ **VE** $\uparrow$ | $\oslash$ **SMAPE** $\downarrow$ |
|---|---|---|---|
| `dorsogna-1k` | Ours (joint, v3) | **0.931**±0.004 | **0.067**±0.002 |
| | Ours (PH-only, v1) | 0.851±0.008 | 0.097±0.005 |
| | PSK [23] | 0.828±0.016 | 0.096±0.006 |
| | Crocker Stacks [57] | 0.746±0.023 | 0.150±0.005 |
| `dorsogna-10k` | Ours (joint, v3) | **0.689**±0.021 | **0.088**±0.004 |
| | Ours (PH-only, v1) | **0.680**±0.025 | **0.090**±0.005 |
| | PSK [23] | 0.647±0.005 | 0.100±0.003 |
| | Crocker Stacks [57] | 0.343±0.016 | 0.145±0.001 |
| `vicsek-10k` | Ours (joint, v3) | **0.576**±0.030 | **0.144**±0.006 |
| | Ours (PH-only, v1) | **0.579**±0.034 | **0.146**±0.006 |
| | PSK [23] | 0.466±0.009 | 0.173±0.003 |
| | Crocker Stacks [57] | 0.345±0.005 | 0.190±0.001 |
| `volex-10k` | Ours (joint, v3) | **0.871**±0.019 | **0.081**±0.006 |
| | Ours (PH-only, v1) | **0.869**±0.018 | **0.082**±0.007 |
| | PSK [23] | 0.509±0.003 | 0.190±0.003 |
| | Crocker Stacks [57] | 0.076±0.019 | 0.292±0.004 |

**Limitation(s).** One obvious limitation in the presented implementation is the reliance on Vietoris-Rips persistent homology of point clouds. In fact, the underlying simplicial complex will become prohibitively large (especially for $H_2$) once we scale up the number of points by, e.g., an order of magnitude. Using an Alpha [22] or a Witness complex [19] might be a viable alternative to mitigate this issue. Similarly, one may explore subsampling strategies for persistent homology, as in [11], and learn a latent ODE from a combination of estimates.

**Societal impact.** The present work mainly deals with an approach to capture the dynamics observed in data representing collective behavior. We assume that this has no direct societal impact, but point out that any application of such a model (aside from synthetic data), e.g., in the context of behavioral studies on living beings or health sciences (e.g., to study the development of cancer cells), should be carefully reviewed. This is especially advisable when drawing conclusions from missing measurements, as these are based on imputations and may be biased.

### Acknowledgments

This work was supported by the Land Salzburg within the EXDIGIT project 20204-WISS/263/6-6022 and projects 0102-F1901166- KZP, 20204-WISS/225/197-2019. M. Uray and S. Huber are supported by the Christian Doppler Research Association (JRC ISIA). *We also thank all reviewers for the valuable feedback during the review process.*

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

# Supplementary material

## A  Simulation settings

As mentioned in Sec. 4.1, we create datasets of dynamic point clouds from simulations. All simulations are run with 200 points for 1,000 steps with a step size of $0.01$, and every 10th step is taken as an observation. Only for the final ablation experiment of Sec. 4.3 (i.e., *Impact of the observation timeframe*), we simulate 20,000 time steps (of the D'Orsogna model) and extract training/testing sequences from this extended timeframe. The model parameters for these simulations are randomly sampled as specified below, and for each sampled parameter tuple, we create one simulation.

To create the `dorsogna-10k` dataset, we vary the following model parameters (see Fig. 3): *overall, we have **four** parameters that need to be predicted.* As macroscopic regimes mainly depend on the ratios $C_r/C_a$ and $l_r/l_a$, cf. [18, Fig. 1], we fix $C_a = l_a = 1$ and sample $C_r = 2^{t_C}$, $l_r = 2^{t_l}$ with uniformly distributed $t_C \sim \mathcal{U}_{[-1,1]}$ and $t_l \sim \mathcal{U}_{[-1.5,0.5]}$. Similarly, we sample $\alpha = 2^{t_\alpha}$ with $t_\alpha \sim \mathcal{U}_{[-2,2]}$ and $m = 2^{t_m}$ with $t_m \sim \mathcal{U}_{[-2,2]}$. The model from the `SiSyPHE` library used to implement this simulation is `AttractionRepulsion`. Note, that the `SiSyPHE` library implements the mass $m$ in terms of the interaction radius parameter $R$, i.e., $m = R^3$ (as we simulate point clouds in 3D).

**Remark 4.** *For comparability (and interpretability of the parameters) to the original D'Orsogna model from [18], we adjusted the* `AttractionRepulsion` *implementation of* `SiSyPHE` *to directly match [18, Eqs. (2) & (3)].*

The `dorsogna-1k` dataset is created by re-running the simulation provided as part of the public (Julia) implementation[3] of [23]. This dataset has ***two** parameters to predict*. Note that in [23], the authors simulated 500 sequences. For our work, we simulated 1,000 to have a larger dataset, but still one magnitude smaller than `dorsogna-10k`, `vicsek-10k` and `volex-10k`. For this dataset, particle masses are $m = 1$, propulsion is $\alpha = 1$, $C_a = l_a = 1$ and $C_r, l_r$ vary uniformly in $[0.1, 2]$, and are selected if the generated point clouds satisfy a certain scale condition, leading to the parameter pairs illustrated in [23, Fig. 6].

For the `vicsek-10k` dataset, we sample $R, c, \nu$ uniformly from $\mathcal{U}_{[0.5,5]}$ and $D$ from $\mathcal{U}_{[0,2]}$. *Overall, this gives **four** parameters that need to be predicted.*

Finally, for the `volex-10k` dataset, we sample ***four** parameters* as follows: $\alpha \sim \mathcal{U}_{[0,2]}$, interaction radii $R \sim \mathcal{U}_{[0,2]}$ and birth/death rates $\lambda_b, \lambda_d \sim \mathcal{U}_{[0,1]}$. However, for the latter two parameters, we discard settings where $\lambda_d > \lambda_b$ as, in this case, death rates are almost impossible to distinguish; also, we discard settings where $\lambda_b \gg \lambda_d$ as the resulting point cloud cardinalities exceed 2000 points during the simulation. The resulting $(\lambda_b, \lambda_d)$ combinations are illustrated in Fig. 5.

---

[3]https://github.com/ldarrick/paths-of-persistence-diagrams

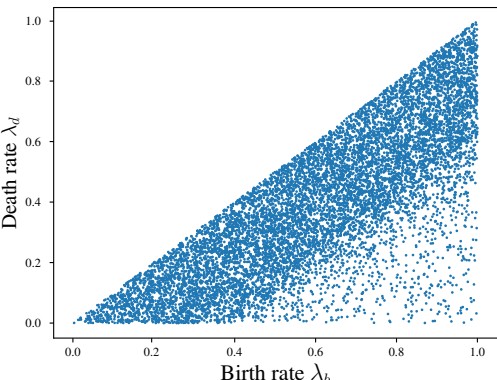

**Figure 5:** Birth rates $\lambda_b$ and death rates $\lambda_d$ used for generating the `volex-10k` dataset.

*For reproducibility, we will release the simulation data publicly.*

# B    Comparison(s) to prior work & Additional ablation experiments

## B.1    Path Signature Kernel (PSK) [23]

For the PSK, we rely on the publicly available implementation in `sktime`[4] [38]. We compute the PSK (truncated at signature depth 3) using our vectorized persistence diagrams per time point as input. For concatenated zero-, one- and two-dimensional persistence diagrams, we use 20-dimensional vectorizations (as specified in Sec. 4.2), yielding 60-dimensional input vectors per time point. The computed kernel is then input to a kernel support vector regressor (kernel-SVR).

Following the experimental protocol in [23, Sec. 7.3], we cross-validate the sliding window embedding (across lags $[1, 2, 3]$) and kernel-SVR hyperparameters on a 20% validation portion of the training data, with hyperparameters selected based on the average MSE (per governing equation parameter $\beta_i$ to be predicted) across 5-folds. Due to the excessive runtime (approx. 8.3 hours per lag on the system specified in Appendix C), we only report results when *all* time points per observation sequence are considered. Hence, *no subsampling* of time points (as is done when evaluating our approach) is performed, and the reported performance can be considered an *optimistic* estimate of what can be achieved with the PSK.

## B.2    Crocker stacks [57]

As mentioned in Sec. 4.4, we compare against *crocker stacks*, introduced in [57], as one of the state-of-the-art approaches. Crocker stacks are an extension to *crocker plots* [52] and constitute a topological summary for time-varying persistence diagrams. For the crocker stacks, we adapted the publicly available implementation of the crocker plots in the `teaspoon`[5] library.

The crocker plot is computed as follows: for each time step, the persistence diagrams are computed up to a scale parameter $\varepsilon$. In discretized steps of the scale parameter, the Betti numbers are computed from the persistence diagrams, which results in a 2D representation (i.e., $\epsilon$ vs. Betti number) for each homology dimension.

The extension to crocker stacks is achieved by adding a third dimension $k$, induced by the smoothing factor $\alpha$. For given steps of $\alpha$, a smoothing operation is applied, i.e., values within a specified distance (smoothing factor) from the diagonal of the persistence diagram are ignored. Hence, for each homology dimension, a crocker stack is a tensor in $\mathbb{R}^3$, with axes corresponding to discretizations of the (1) scale parameter $\varepsilon \in [0, \infty)$, the (2) time $t \in [0, T]$, and the (3) smoothing factor $\alpha \in [0, \infty)$.

We note that in [57], the authors set the scale parameter to $\delta = 0.35$, as they use data normalized to the range of 0 to 1. We *did not* scale the data, however, we adjusted the scale parameter correspondingly:

---

[4]https://github.com/sktime/sktime
[5]https://teaspoontda.github.io/teaspoon

for all experiments, we set the scale parameter to $\delta = {}^{1}\!/_{3} \cdot \mathrm{maxPers}(\mathrm{dgm}_k(\mathrm{Rips}(\mathcal{P})))$, where $\mathcal{P}$ denotes the point cloud, and $\mathrm{maxPers}(\mathrm{dgm}_k(\mathrm{Rips}(\mathcal{P})))$ is the maximum persistence obtained from all point clouds in an observed sequence.

**Hyperparameter choices.** We made the following hyperparameter choices: (1) the Vietoris-Rips filtration is considered up to $\delta$ (see above), where the computations are discretized with 25 equally spaced values in $[0, \varepsilon]$; (2) the smoothing values are discretized with 18 steps equally spaced in $[0, 0.5 \cdot \mathrm{maxPers}]$.

Eventually, the crocker stacks per homology dimension are vectorized (i.e., the tensor is flattened) and concatenated into a single vector per observation sequence. These vectorizations are then input to a linear support vector regressor (SVR). For each of the (varied) parameters in the governing equation of Fig. 3, a *separate* SVR is trained and evaluated. Hyperparameters of the SVR and the usage of preprocessing steps (feature scaling) are tuned using Bayesian optimization. Each parameter configuration is evaluated using a 5-fold cross-validation, and the best configuration is then used to train the final model on the full dataset.

### B.3 Additional ablation experiments

To assess whether homology dimensions $> 0$ are beneficial to the downstream regression task, we experimented on `dorsogna-1k`. We find that when using our approach with $H_0$-features only, the regression quality drops. When additionally including $H_2$-features (i.e., $H_0$, $H_1$ and $H_2$), the situation is less clear, as the results are not noticeably different. Quantitative results can be found in Tbl. 5 below.

**Table 5:** Quantitative assessment of the impact of including higher-dimensional homology features on downstream regression performance, evaluated on `dorsogna-1k`.

|  | $\oslash$ **VE** $\uparrow$ | $\oslash$ **SMAPE** $\downarrow$ |
| --- | --- | --- |
| Ours (PH-only, v1; $H_0$) | $0.819 \pm 0.015$ | $0.101 \pm 0.003$ |
| Ours (PH-only, v1; $H_0, H_1$) | $0.844 \pm 0.021$ | $0.098 \pm 0.007$ |
| Ours (PH-only, v1; $H_0, H_1, H_2$) | $0.846 \pm 0.011$ | $0.097 \pm 0.005$ |

## C Computational resources

All experiments were run on an Ubuntu Linux system (22.04), running kernel 5.15.0-100-generic, with 34 Intel® Core™ i9-10980XE CPU @ 3.00GHz cores, 128 GB of main memory, and two NVIDIA GeForce RTX 3090 GPUs.

## D Runtime analysis

In the following, we present a runtime breakdown of the pre-processing steps (i.e., Vietoris-Rips persistent homology (PH) computation, and the vectorization of persistence diagrams), as well as a runtime comparison to prior work (*PSK* and *crocker stacks*) and our baseline approach. Runtime is measured on the system listed above, using our publicly-available reference implementation.

*At this point, it is also worth highlighting that our pre-processing step, i.e., PH computation & vectorization, is trivially parallelizable and can easily be distributed across multiple CPUs/GPUs if needed.*

**Vietoris-Rips PH computation.** In Tbl. 6, we list wall clock time measurements (using Ripser++ [59] on one GPU) per point cloud. In particular, the table lists runtime for computing PH of dimension zero and one (i.e., $H_0$ and $H_1$), as well as PH of dimension zero, one and two (i.e., $H_0, H_1$ and $H_2$). All point clouds in this experiment are of size 200 in $\mathbb{R}^3$.

**Persistence diagram vectorization.** Persistence diagram (PD) vectorization can essentially be broken down into two steps: (1) parameter fitting for the structure elements of [26] and (2) mapping PDs to vector representations using those structure elements. Tbl. 7 lists the runtime for both steps on the `dorsogna-1k` data when vectorizing zero-, and one-dimensional persistence diagrams. Throughout

**Table 6:** Runtime comparison for *PH computation* for different homology dimensions. Reported is the average runtime per point cloud (in seconds), and the overall runtime (in seconds) estimated from this average by multiplying by the number of processed point clouds on `dorsogna-1k`.

| | ⊘ **Time per point cloud** | **Overall** |
|---|---|---|
| PH computation $(H_0, H_1)$ | 0.018 s | 1800 s |
| PH computation $(H_0, H_1, H_2)$ | 0.330 s | 33000 s |

all of our experiments, parameter fitting for the structure elements is done by first collecting all persistence diagrams for each homology dimension and then running $k$-means++ clustering on 50,000 points uniformly sampled points from those diagrams.

**Table 7:** Runtime breakdown of *PD vectorization* (per diagram and overall for $H_0$ and $H_1$ on `dorsogna-1k`), split into (**Step 1**) time spent for fitting the parameters of the *exponential structure elements* from [26], and (**Step 2**) actually mapping PDs to vector representations.

| | ⊘ **Time per diagram** | **Overall** |
|---|---|---|
| **Step 1**: Parameter fitting | n/a | 17 s |
| **Step 2**: Mapping PDs to vectors | 9.4e-05 | 18 s |

**Comparison to prior work.** Next, we compare the runtime of our approach (here, variant v1 from Fig. 2) to prior work that uses persistence diagrams as input. In particular, we compare against the PSK method from [23] and crocker stacks [57].

In Tbl. 8, we list the overall *training time* (on `dorsonga-1k`), where runtime for pre-processing (see above) is excluded from these measurements. Also, PSK and crocker stacks timings do include hyperparameter optimization, as suggested in the corresponding references. Importantly, this is not optional but required to obtain decent performance with respect to EV and SMAPE. Notably, for the PSK approach, kernel computation scales quadrati-

**Table 8:** Training time comparison to prior work (on `dorsogna-1k`).

| | **Training time** |
|---|---|
| Crocker Stacks | 24600 s |
| PSK | 646 s |
| **Ours** (PH-only, v1) | 190 s |

cally with the number of sequences, and kernel-SVR training takes time somewhere between quadratic and cubic. Hence, scaling up the number of training sequences quickly becomes computationally prohibitive, especially in light of the required hyperparameter tuning. Finding suitable hyperparameters is also the main bottleneck for crocker stacks (which rely on a linear SVR).

**Comparison to baseline model.** Finally, in Tbl. 9, we present a training time comparison to our *baseline model* which does not explicitly model any dynamics via a latent ODE (denoted as w/o dynamics). In this experiment, the training protocol remains unchanged, and we vary the type of input data, i.e., from using vectorized persistence diagrams only ($H_0$ and $H_1$) to using a combination of PointNet++ features *and* vectorized persistence diagrams. Importantly, as remarked in the main part of the manuscript, the baseline models (PH-only, v1) and (PH+PointNet++, v3) already yields strong performance across all parameter prediction problems.

**Table 9:** Training time comparison (on `dorsogna-1k`) of our approach vs. the baseline model (w/o dynamics), using the same input data.

| | **Training time** |
|---|---|
| **Ours** (PH-only, v1) | 190 s |
| **Ours** (PointNet++, v2) | 3780 s |
| **Ours** (PH+PointNet++, v3) | 4100 s |
| Baseline (w/o dynamics; PH-only, v1) | 50 s |
| Baseline (w/o dynamics; PointNet++, v2) | 525 s |
| Baseline (w/o dynamics; PH+PointNet++, v3) | 600 s |

