# OpenReview forum: "Neural Persistence Dynamics"
_NeurIPS.cc/2024/Conference — NeurIPS 2024 poster_

### Official Review · Reviewer_HnKN · 2024-06-20

**Soundness:** 2
**Presentation:** 2
**Contribution:** 2
**Rating:** 5
**Confidence:** 3

**Summary:**

This work presents a novel approach to infer the parameters of governing equations describing the collective behavior of systems like point clouds. It leverages persistent homology to capture the topological features of the system's state. These features are then modeled using a Latent ODE system, capturing the temporal evolution of the system.  By analyzing the latent dynamics, the method can identify and regress the parameters of the underlying governing equations (e.g., PDE) that govern the system's behavior.

**Strengths:**

- This work proposes a novel method based on persistent homology (PH) to infer the parameters of governing equations describing the collective behavior of systems like point clouds.
- A novel application in combining persistent homology with inverse problems.
- The paper discusses various aspects of combining PH with modeling temporal correlations, giving an in-depth insight into the field.

**Weaknesses:**

- This work resonates closely with inverse problems for dynamical systems, where several works have been conducted to infer the initial parameters or parameters of interest based on observed data. Some of them are:

           1. Learning to Solve PDE-constrained Inverse Problems with Graph Networks (https://arxiv.org/pdf/2206.00711)
           2. Fully probabilistic deep models for forward and inverse problems in parametric PDEs
           3. Invertible Fourier Neural Operators for Tackling Both Forward and Inverse Problems (https://arxiv.org/pdf/2402.11722)

A comparison with these methods will strengthen the aspect of this work, as these methods do not explicitly model the topological features.

- I think the paper should include some additional discussion about the computational limitations of the method. While these are briefly discussed at a high level, a more quantitative analysis (e.g., comparing the effective run times and computational complexity) seems essential to give the reader a better idea of the overhead incurred when using this method.
- Despite the capability of their model to incorporate persistent homology, enabling the use of topological features, they only utilize classical persistent homology via Rips filtrations. The potential advantages of this feature are not adequately explored in the model.

**Questions:**

- Does the model output a single point estimate for the parameters of interest or a corresponding distribution (mean and std)?
- Can authors describe the regression objective w.r.t to underlying parameters of interest, as it is unclear from the paper, maybe including the correct equations they use?
- Since PH does not account for the temporal changes in the point cloud,
- How is the work different from this https://arxiv.org/abs/2406.03164? It seems like it can be subsumed as one can define a graph out of point clouds and operate on them.
- Does the fitting of the ODE system deal with stiffness as you are fitting the topological features via an ODE, which can be non-smooth in between, leading to inefficient modeling?

**Limitations:**

See questions and weaknesses.

---

> ### Author Rebuttal · Authors · 2024-08-06
>
> For better readability, we restate your comments/questions in *italic*, our response(s) are marked by &#9654;
>
> *This work resonates closely with inverse problems for dynamical systems, where several works have been conducted to infer the initial parameters ...*
>
> ▶ Currently, we discuss related work on the inverse problem, but only in the context of models of collective behavior. We thank the reviewer for suggesting a broader perspective and comparing with prior learning-based works on inverse problems for dynamical systems in general, including the three listed ones. This will certainly be a good addition to the manuscript.
>
> Regarding refs 1-3, we note that the underlying differential equations (i.e., classic PDEs) differ from the multi-particle systems we consider. In our case, forces on each particle depend on its relative position towards all other (or its neighboring) particles, resulting in strong couplings between individuals. For specific systems (Volume exclusion), it is possible to derive parametric PDEs that describe the asymptotics of infinitely many particles (Oelschläger, 1990), and ideas from references 1-3 might be applicable. However, in general, the number $N$ of particles influences the dynamics. E.g., for the D'Orsogna model, particle distances can converge to 0 as $N\to \infty$.
>
> *I think the paper should include some additional discussion about the computational limitations of the method ...*
>
> &#9654; We agree; please see our *General Response* section.
>
> *Despite the capability of their model to incorporate persistent homology, enabling the use of topological features, they only utilize classical persistent homology ...*
>
> &#9654; Vietoris-Rips persistent homology is the de-facto workhorse for point cloud data. While one could use other geometric complexes (e.g., Alpha complexes as mentioned by reviewer **dbj5**) or different vectorization strategies (ATOL, persistence images, etc.), we decided to work with the most prevalent tools, but acknowledge that other choices are possible. Also, using Vietoris-Rips PH facilitates straightforward comparisons to previous topology-related works on the same problem, such as Crocker Stacks or the PSK approach.
>
> Importantly, our goal was not to assess the impact of different design choices but to introduce a latent dynamical model based on topological per-time-point descriptors for the inverse problem at hand. We are convinced that future work will explore different variants of our approach that may be more suitable in certain situations (e.g., using LSTMs in case of discrete dynamical systems, see **83FG**).
>
> *Does the model output a single point estimate for the parameters of interest or a corresponding distribution (mean and std)?*
>
> &#9654; At the moment, all models output *point estimates* of the sought-for parameters. However, one could easily sample $q_{\\theta}$, integrate forward, and obtain a distribution of estimates. We will add tables to the appendix that report an estimated mean & std. dev. from this sampling.
>
> *Can authors describe the regression objective w.r.t to underlying parameters of interest, ...?*
>
> &#9654; The objective is to predict the simulation parameters that led to a particular realization of a point cloud sequence. Take the PH-only variant `v1` of Fig. 1, for instance: input is a sequence of vectorized persistence diagrams $v_{\\tau_0}, \\ldots, v_{\\tau_n}$. The encoder $Enc_\\theta$ yields parameters of the approx. posterior $q_\\theta$ from which we sample and integrate the latent ODE forward in time to get latent states along the trajectory; this latent state sequence is then fed through $Enc_\\alpha$ which summarizes the sequences into a vector and linearly maps the latter to simulation parameter estimates
> $\\hat\\beta = (\hat{\beta}\_1, \ldots, \hat\beta\_P)$. During training, we minimize the mean-squared-error (MSE) between the predictions and the ground-truth simulation parameters, implicitly making a Gaussian noise assumption. One could write the regression model as
> $$\\hat\\beta = \\phi(v_{\\tau_0}, \\ldots, v_{\\tau_0}) + \epsilon, \epsilon \sim \\mathcal N(0,\sigma I)$$
> where $\phi$ subsumes all steps above.
>
> *Since PH does not account for the temporal changes in the point cloud, how is the work different from this https://arxiv.org/abs/2406.03164? ...*
>
> &#9654; As correctly pointed out, we do not track topological features on the level of *individual points* in a persistence diagram but on the level of *vectorized persistence diagrams*. The crucial difference to the arXiv paper is that the integration of neural ODEs is on the *message passing level* of a GNN for a *fixed* graph. In our context, this means that the graph object needs to be such that each vertex corresponds to the same particle at all times. However, such correspondences are unknown or hard to obtain in practice. In fact, in our work, we deliberately avoid this "tracking" step.
>
> *Does the fitting of the ODE system deal with stiffness as you are fitting the topological features via an ODE, which can be non-smooth in between, leading to inefficient modeling?*
>
> &#9654; This is an interesting point! However, we *do not* model the dynamics directly in the space of persistence diagrams but rather in a latent space learned from diagram vectorizations. This modeling could be inefficient to some extent, yes, but the latent ODE approach seeks to learn the most suitable latent space for the task at hand, which is to minimize prediction error for the model parameters (via the MSE) and to minimize the reconstruction error for the vectorized persistence diagrams (as part of the ELBO). This strategy is common in the literature, as, e.g., seen in the PhysioNet 2012 experiments of (Rubanova et al., 2019), where variables include binary indicators (e.g., of whether mechanical ventilation is used at available time points).

---

> > ### Author Response · Authors · 2024-08-13
> >
> > We would like to kindly ask, at the end of this rebuttal phase, whether our response addressed your questions/concerns or whether we can provide any further clarifications.

---

### Official Review · Reviewer_83FG · 2024-06-23

**Soundness:** 2
**Presentation:** 3
**Contribution:** 3
**Rating:** 6
**Confidence:** 3

**Summary:**

The paper addresses the challenge of predicting the specific parameters of models yielding point cloud dynamical systems known only partially from a set of observations in different time steps. This is achieved by leveraging information about the evolution of persistent homology vectorizations of the observed point clouds at different time steps. Particularly, the paper uses a specific, previously published, vectorization that is Lipschitz with respect to the standard point set Wasserstein distance. The evolution of the topological vectorization for the different point cloud dynamical systems is assumed to be governed by latent dynamic processes, which are approximated via a neural ODE and used to infer, given a set of observations of an evolving point cloud, the model parameters that produce these observations. The proposed baseline method is tested and benchmarked across three different scenarios, demonstrating the utility of its components and showing a significant superiority over the leading state-of-the-art methods.

**Strengths:**

**Originality**: To the best of my knowledge, the idea of modeling the dynamics of vectorized persistence diagrams via a continuous latent variable model is novel and powerful.

**Significance**: As stated and demonstrated in the paper, neural persistence dynamics, and the evolution of point clouds over time seen as a group rather than individual points, provide significant insights into the dynamical systems governing the behavior of the individual points. This approach has great potential in studying real-world problems, especially in "natural systems" where the interaction of individuals as a group is key to understanding their behavior (e.g., flocks of birds, insects, fish, cells in an organism).
Specifically related to neural networks, the method proposed in this paper has the potential to improve the analysis of the dynamics of data going through the different layers of a neural network, as done in a more simplistic way in [1] and [2], where contradictions were found that this project might help resolve. Additionally, the method's significant superiority compared to other approaches for predicting the parameters of dynamical systems from evolving point clouds suggests that this is indeed a promising direction for further research in the study of evolving topology and in the aforementioned areas.

**Clarity and Quality**: The paper is generally well-written, with a few exceptions that I will address in the weaknesses section. The comprehensive literature review significantly enhances the quality of the manuscript. Figure 1 and 2 are particularly clarifying when reading the text.

[1] Naitzat, Gregory, Andrey Zhitnikov, and Lek-Heng Lim. "Topology of deep neural networks." Journal of Machine Learning Research 21.184 (2020): 1-40.

[2] Wheeler, Matthew, Jose Bouza, and Peter Bubenik. "Activation landscapes as a topological summary of neural network performance." 2021 IEEE International Conference on Big Data (Big Data). IEEE, 2021.

**Weaknesses:**

- I think that the sentence in line 32 "... due to missing correspondences between individuals..." is hard to read. Which missing correspondences?

-Table 1 lacks context and is duplicated in Table 2. The experiments of this table are not explained and metrics are not justified. I think a more qualitative explanation may be better than the quantitative explanation given in the introduction. Alternatively, providing more information about the experiments would help, but it could lead to information duplicity, which might be undesirable.

- The proposed approach does not track of the evolution of individual topological features of the persistence diagram. This could be important in scenarios requiring fine-grained topological information.

- The authors claim that their method is more efficient than other methods, but computation times are not reported in the main text.

- In the persistent homology paragraph (starting at line 154), the authors state that there is a decomposition in persistence diagrams (births and deaths) when using abelian groups in the chain complexes. As far as I know, this is not a trivial property and does not happen always [Theorem 2, reference 3] for abelian groups. The usual theorem applies to vector spaces.

- Only one vectorization method is used, without significant justification. I would have appreciated a more detailed ablation on this point.

- Maybe I'm wrong, but when I access the arxiv version of reference [23] in your paper, Table 4 contains execution times, and not SOTA results.

[3] Jiajie Luo, , Gregory Henselman-Petrusek. "Interval Decomposition of Persistence Modules over a Principal Ideal Domain." (2023).

**Questions:**

- Why do you use $R^2$? I thought that it was a misleading metric [4]. This is my main concern, and the core reason why the score is a borderline accept.
- Can your approach be used for discrete dynamical systems (where time is discrete and not continuous?)
- Since the method is very general and can work with any kind of vectorizations, not just topological ones, is there a way to regularize or add explicit bias during training to inform the training process/neural ODE that the representations are topological?

[4] Li J (2017) Assessing the accuracy of predictive models for numerical data: Not r nor r2, why not? Then what?. PLOS ONE 12(8): e0183250.

**Limitations:**

Some limitations have been addressed, although I would appreciate a better discussion, for example, adding that the evolution of individual topological features cannot be studied.

---

> ### Author Rebuttal · Authors · 2024-08-06
>
> For better readability, we restate your comments/questions in *italic*, our response(s) are marked by &#9654;
>
> *I think that the sentence in line 32 "... due to missing correspondences between individuals..." is hard to read. Which missing correspondences?*
>
> &#9654; By "correspondences" we mean that many previous works rely on being able to unambiguously identify particles across point clouds over time. We will clarify this unfortunate formulation.
>
> *The proposed approach does not track of the evolution of individual topological features of the persistence diagram. This could be important in scenarios requiring fine-grained topological information.*
>
> &#9654; Indeed, our method does not retain information about how individual homology classes evolve. We model the evolution of features *on the level of persistence diagram vectorizations*, not on the level of individual points in the persistence diagram(s). Persistence vineyards would allow us to do this, but they are (i) hard to vectorize for ML tasks, (ii) computationally expensive to compute, and (iii) often infeasible to obtain without particle correspondences over time. Following your suggestion, we will extend our discussion to include these points.
>
> *The authors claim that their method is more efficient than other methods, but computation times are not reported in the main text.*
>
> &#9654; Please see our runtime analysis in the *General Response* section.
>
> *In the persistent homology paragraph (starting at line 154), the authors state that there is a decomposition in persistence diagrams (births and deaths) when using abelian groups ...*
>
> &#9654; When sketching persistent homology in Sec. 3, our writing unintentionally and wrongly suggests that birth/death times (interval compositions) are well-defined for homology with coefficients in arbitrary Abelian groups, e.g., in the presence of torsion. We will clarify this part; thank you for pointing it out. We use coefficients in the field $\\mathbb{Z}/2\\mathbb{Z}$ (as is usual).
>
> *Only one vectorization method is used, without significant justification. I would have appreciated a more detailed ablation on this point.*
>
> &#9654; Our contribution is not within the particular realization of our core idea, but in the surprising insight that one can obtain remarkable predictive performance from capturing the dynamics of vectorized topological summaries per time point, without tracking particles, or without computationally-expensive summaries of time-varying persistence. On could, however, use other vectorization techniques; we experimented with ATOL vectorizations (which are conceptually very close to ours) and persistence images, but only observed minor changes in predictive performance. Nevertheless, ATOL vectorizations are not stable, and persistence images can be hard to parametrize (and discretize) without obtaining (very) high-dimensional descriptors.
>
> *Maybe I'm wrong, but when I access the arxiv version of reference [23] in your paper, Table 4 contains execution times, and not SOTA results.*
>
> &#9654; Thank you for pointing this out! Table 4 is meant to refer to the table in the submitted manuscript, not in the referenced work. We will rephrase our formulation.
>
> *Why do you use 𝑅2 ? I thought that it was a misleading metric [4]. This is my main concern, and the core reason why the score is a borderline accept.*
>
> &#9654; We agree, $R^2$ has known flaws, much like other metrics to assess regression performance, yet it is easy to interpret. As a summary impression across *all* predicted parameters, MSE and RMSE cannot be applied due to differing ranges of simulation parameters. We used $R^2$ and SMAPE to have two different evaluation scores. Our *joint* variant (v3) outperforms the state-of-the-art in both of these scores across all datasets, and even our PH-only variant (v1) does so on 3 out of 4 datasets.
>
> To further reduce evaluation bias, we will add the explained variance (EV) as suggested in your ref [4] to all tables, and also list the RMSE per parameter in our appendix. Below is an example for the `dorsogna-1k` data (parameters $C,l$ are as in (Guisti et al., 2023)):
>
> | | $R^2$ $\uparrow$  | SMAPE $\downarrow$ | RMSE ($C$) $\downarrow$ | RMSE ($l$) $\downarrow$ | EV $\uparrow$ |
> | ---|---|---|---|---|---|
> | Ours (joint, v3) | 0.930 $\pm$ 0.003 | 0.068 $\\pm$ 0.003 | 0.080 $\\pm$ 0.005 | 0.190 $\\pm$ 0.004 | 0.931 $\\pm$ 0.004 |
> | Ours (PointNet++, v2) | 0.814 $\\pm$ 0.032 | 0.132 $\\pm$ 0.018 | 0.178 $\\pm$ 0.017 | 0.242 $\\pm$ 0.020 | 0.816 $\\pm$ 0.003 |
> | Ours (PH-only, v1)| 0.846 $\\pm$ 0.011 | 0.097 $\\pm$ 0.005 | 0.116 $\\pm$ 0.004 | 0.275 $\\pm$ 0.006 | 0.851 $\\pm$ 0.008 |
> | PSK | 0.816 $\pm$ 0.015 | 0.096 $\\pm$ 0.006  | 0.112 $\\pm$ 0.010 | 0.305 $\\pm$ 0.013 | 0.819 $\\pm$ 0.016 |
> | Crocker Stacks | 0.743 $\\pm$ 0.083 | 0.150 $\\pm$ 0.005 | 0.156 $\\pm$ 0.007 | 0.331 $\\pm$ 0.011 | 0.746 $\\pm$ 0.023|
>
> *Can your approach be used for discrete dynamical systems (where time is discrete and not continuous?)*
>
> &#9654; We are confident that our method can be used with discrete systems. However, some components of our architecture may need to be adjusted; e.g., switching from neural ODEs for modeling latent dynamics to, say, LSTMs or GRUs. Thank you for this comment, we will add a remark!
>
> *Since the method is very general and can work with any kind of vectorizations, not just topological ones, is there a way to regularize or add explicit bias during training ....?*
>
> &#9654; If we understand your question correctly, you are asking whether some form of topological prior could be used to inform the neural ODE training process. This is an interesting question, and while we have not experimented with topological regularization so far, it might be possible to enforce specific topological/geometrical properties through an appropriate loss (on persistence diagrams) and differentiating through the PH computation (as done in prior works).

---

> > ### Comment · Reviewer_83FG · 2024-08-09
> >
> > Really good answer, thank you very much. You addressed all my comments. Although I think it could have been really interesting to add individual tracking of persistent homology features (e.g., using vineyards, as you propose), I think the paper should be accepted for the conference. Also, I would love to see a continuation of this paper addressing discrete dynamical systems. I know several problems where this could be used successfully. In general, I'm really excited about this particular direction of applied TDA, so I'm increasing my score to 6. Congratulations and thank you very much again!

---

### Official Review · Reviewer_dbj5 · 2024-06-28

**Soundness:** 3
**Presentation:** 2
**Contribution:** 3
**Rating:** 5
**Confidence:** 4

**Summary:**

This paper considers the problem of learning some parameters $\theta$ in---roughly speaking---a dynamical system of the form $\dot X = \phi(X, \theta)$, where $X \in \mathbb{R}^{n \times d}$ (with typically $d=3$), from an observed time-discrete trajectory $X(\tau_0),\dots,X(\tau_N)$.

The idea conveyed by this paper is that instead of tracking the dynamic of the whole point cloud ($n$ can be quite large) to infer the parameter $\theta$, it may be somewhat sufficient to look at a "featurization" of it---here, of a topological nature, yielding a sequence of vectors $v_{\tau_0},\dots,v_{\tau_n}$ that somewhat summarize the dynamic while being (hopefully) sufficient to infer the parameters $\theta$.

The parameters of the system are then learned by existing techniques, essentially by using a low-dimensional latent dynamic $(z_t)_t$ modeled by a neural ODE and then connected through the observed $(v_t)_t$ by encoder-decoder; training being performed using ELBO maximization.

The experiments are conducted on synthetic data (from rather sophisticated dynamical systems), containing an extensive ablation study, and showcasing the usefulness of the proposed method on the considered models.

**Strengths:**

The proposed use-case of Topological Data Analysis in the context of dynamical systems feels quite convincing and refreshing.

The methodology, at a global picture level, is fairly clear.

The performance showcased by the experimental results, supported by an extensive ablation study, seems to robustly support the usefulness of the method.

**Weaknesses:**

My main concern is about the clarity of the paper in the following sense: at the global picture level, the paper feels very clear and convincing. However, I think that it is not clear at the low-level scale, in that I feel completely unable to implement the method based on what is described in the (main) paper.
For, the PD vectorizations are never described precisely, the final loss function (ELBO + regression) is only informally discussed in lines 226-234, I actually do not exactly understand how the parameters of the systems are encoded in the models, between which variables the $R^2$ coefficient (line 268) is computed (I infer that it is the parameters of the models wrt the one inferred at training time), etc.

I understand that code and so on will be delivered and that the method will be usable by others if needed, but I believe that the impact of the paper would be strengthen if one could understand what is done _at the technical level_ (i.e., be able to roughly implement it) after reading it.

Adding a minimalist proof-of-concept experiment, with a trivial dynamical system like $\dot x = \theta x$ or something like that, may possibly be helpful (or not).

**Questions:**

1. Related to my comments above, could you make explicit the way the parameters of the systems are encoded and learn? Say that I consider the Volume exclusion model with parameters $(\alpha, R, \lambda_b, \lambda_d)$. How are these parameters actually estimated once $\texttt{Enc}_\alpha, \texttt{Enc}_\theta, \texttt{Dec}_\gamma$ are trained with the ELBO+regression rule? How are the $\hat{\beta}_t$ (showcased in Figure 3) used afterward?

2. (more genuine question out of curiosity) In some sense, the seminal point cloud $X$ contains implicitly all the possible topological information one could extract using persistent homology. It also seems that PointNet-like architectures can extract this topological information---see for instance the RipsNet model proposed in [1]. So in some sense, this suggests that turning the observed $X(\tau)$ into diagrams is "the way to go" but PointNet fails to learn that. Do you agree with this statement? If so, do you have any intuition on why?

3. Still related to the above, the paper mentions that it uses PD with homology dimension 0, 1 and 2. Did you run the experiments using $H_0$ alone? If the performance significantly decreases, that would be a nice example where non-trivial topology is useful (as frustrating as it can be, I often observed that $H_0$ alone, i.e. looking mostly at the distribution of pairwise distances, was sufficient to reach good performances). Note: please do not feel obliged to run new experiments during the rebuttal period! This is just a genuine question.

4. In the conclusion, you mention that computing $H_2$-PD with Rips filtration is quickly computationally expensive (I completely agree), and propose to use alpha-filtration instead. Is there any difficulty to do so right now? (Alpha filtrations are readily implemented in several Tda-libraries according to https://cat-list.github.io/ ).


Ref :

[1] 	RipsNet: a general architecture for fast and robust estimation of the persistent homology of point clouds, de Surrel et al.

---

> ### Author Rebuttal · Authors · 2024-08-06
>
> For better readability, we restate your comments/questions in *italic*, our response(s) are marked by &#9654;
>
> **ad Clarity:** We tried to balance the presentation of our conceptual idea vs. the technical detail of its particular realization. We can provide more detail on the ELBO objective and the added regression loss in the appendix. The same holds for the PD vectorization technique we use. As both parts appeared separately in prior work, we sacrificed technical detail for those parts in favor of a more comprehensive conceptual perspective. We will add these details to the appendix to facilitate reproducibility, as you suggested, including pointers to the respective parts in the source code.
>
> Further, for **training**, the regression loss is the MSE between the parameter estimates and the ground truth simulation parameters. One may also use $R^2$ (or any other reasonable objective) during training, but we found the MSE to already yield good results. During **inference/evaluation** (on held-out testing data), we report the $R^2$ and SMAPE computed from predictions vs. ground truth values, just as you correctly inferred.
>
> *Related to my comments above, could you make explicit the way the parameters of the systems are encoded and learned? Say that I consider the Volume exclusion model with parameters. How are these parameters actually estimated once $Enc_\\alpha, Enc_\\theta, Dec_\\gamma$ are trained with the ELBO + regression rule. How are the $\hat{\beta}_t$ (showcased in Figure 3) used afterward?*
>
> &#9654; Inferring the simulation model parameters (e.g., for the Volume Exclusion model) is done as follows: For a given sequence of point clouds (at time points $\tau_0, \\dots,\\tau_n$), one first pre-computes Vietoris-Rips persistent homology (e.g., for $H_0, H_1, H_2$) per available time point and vectorizes the diagrams. This yields a sequence of vectors, i.e., one vector per available time point. This sequence is then fed to the encoder $Enc_\theta$ which outputs a parametrization for the approximate posterior $q_\\theta(z_{t_0}|\\{ v_{\tau_i} \\})$. Upon sampling from $q_\\theta$, we get an initial latent state $z_{t_0}$ and integrate the latent ODE forward in time to $t_n$. $Enc_\\alpha$ then summarizes this sequence of latent states and linearly maps it to a vector of *simulation parameter estimates*
> $\beta_1, \\ldots, \beta_P$. (For Volume Exclusion, we have four simulation parameters, so $P=4$).
>
> *(more genuine question out of curiosity) In some sense, the seminal point cloud contains implicitly all the possible topological information one could extract using persistent homology. It also seems that PointNet-like architectures ...*
>
> &#9654; An important point in this context is that RipsNet is *trained* (in a supervised manner) to predict precomputed *vectorizations* of persistence diagrams, not the persistence diagrams themselves. This means the desired vectorized diagram is known in advance and can be used in a loss function. Clearly, this strategy is designed to guide the network towards capturing topological information. Instead, in our experiments, the PointNet++ *directly* predicts the parameterization of the simulation; hence, it would have to learn topological information implicitly (as there is no external guidance towards this goal). The only guidance given is a loss wrt. the predicted simulation parameters. It might, however, be possible to replace our full pre-computation step (for Vietoris-Rips PH) with a pre-trained RipsNet (i.e., pre-trained on point clouds from such simulation experiments). We did not explore this direction here, but we greatly appreciate the comment.
>
> *Still related to the above, the paper mentions that it uses PD with homology dimension 0, 1 and 2. Did you run the experiments using $H_0$ alone? If the performance significantly decreases, that would be a nice example where non-trivial topology is useful (as frustrating as it can be, I often observed that $H_0$ alone, i.e. looking mostly at the distribution ...*
>
> &#9654; Yes, initially, we experimented with $H_0$-only and found that prediction performance drops in that case. The situation is less clear when also including $H_2$ features, where the results are more mixed and often not noticeably different. From a more quantitative perspective, below you can find a table for the `dorsogna-1k` experiment comparing $H_0$ vs. $(H_0, H_1)$ vs. $(H_0, H_1, H_2)$ for our approach (i.e., the `v1` variant from Fig. 2):
>
> |   | $R^2$ $\uparrow$  | SMAPE $\downarrow$ |
> |---|---|---|
> | Ours (PH-only, v1, $H_0$)               | 81.9 $\pm$ 0.015  |  0.101 $\pm$ 0.003 |
> | Ours (PH-only, v1, $H_0, H_1$)          | 84.4 $\pm$ 0.021  |  0.098 $\pm$ 0.007 |
> | Ours (PH-only, v1, $H_0, H_1, H_2$)     | 84.6 $\pm$ 0.011  |  0.097 $\pm$ 0.005 |
>
> *In the conclusion, you mention that computing $H_2$-PD with Rips filtration is quickly computationally expensive (I completely agree), and propose to use alpha-filtration instead. Is there any difficulty to do so right now? (Alpha filtrations are readily implemented in several Tda-libraries according to https://cat-list.github.io/ ).*
>
> &#9654; There is no inherent limitation to switching out Vietoris-Rips complexes for Alpha complexes. However, we did find in a preliminary experiment that different implementations of Alpha complexes (some of which are in the link you provided) surprisingly yield different diagrams, and without any further in-depth investigation refrained from using the latter in our experiments. Nevertheless, switching from VR to Alpha complexes would, as you suggested, significantly improve the performance of the preprocessing step in which we compute PH for different dimensions.

---

> > ### Comment · Reviewer_dbj5 · 2024-08-08
> > **Thanks**
> >
> > Thank you for taking time answering my review, giving experimental details and clarifying the content of the paper.
> >
> > I like that $H_1 + H_0 > H_0$, that's somewhat of a "good new" for TDA :-)
> >
> > I still need some time to read other reviews / comments and discuss with other reviewers to make my mind clear but I feel more positive about this work.

---

> > > ### Author Response · Authors · 2024-08-13
> > >
> > > We wanted to kindly ask, at the end of this rebuttal phase, whether the other reviewer's comments/answers to our response have helped in your deliberation or whether we can provide some further "short-term" answers to any remaining open issues.

---

### Official Review · Reviewer_bRy6 · 2024-07-09

**Soundness:** 3
**Presentation:** 3
**Contribution:** 2
**Rating:** 6
**Confidence:** 3

**Summary:**

This work considers the problem of learning the latent, continuous-time dynamics underlying time-evolving point clouds. To solve it, it leverages previous work on the persistent homology of point clouds and their vectorization, as well as the PointNet++ network, to obtain static representations of the point clouds at a set of observation times. It then makes use of the LatentODE framework of Rubanova et al. (2019) to infer a continuous-time representation encoding the observed representations.

The authors employ the learned continuous-time representations to tackle the inverse problem of regressing the ground-truth parameters of a set of governing equations from which the point cloud observations were simulated. They show their model outperforms two recent baselines. They also empirically demonstrate that (i) the deep representations obtained via PointNet++ are complementary to those obtained via persistent homology; that (ii) actually modelling the continuous-time dynamics helps with the regression task; and that (iii) the regression tasks increases its complexity with that of the initial condition in the simulation.

*References:*
- Latent ODEs for Irregularly-Sampled Time Series. Rubanova et al. (2019)

**Strengths:**

The paper is very well written, and provides enough information for a reader not versed in the use of persistent homologies for point cloud summarization. The authors justify their methodology, viz. the encoding of vectorized persistent diagrams into a continuous-time latent process, by using stability arguments of previous work. Their argumentation reads well and is convincing.

Another strenght is that, besides empirically demonstrating that their method outperforms two recent baselines, the authors additionally perform reasonable ablation studies that further justify their methodology.

**Weaknesses:**

The main contribution of this work could be read as a direct application of the Latent ODE framework of Rubanova et al. (2019) to the problem of time-evolving point clouds. It can be seen as incremental too, for it leverages stablished methods for point cloud representations. One could therefore argue that, despite its merits, the paper would better fit a conference or journal with a more applied character.

It is also not clear how the method would perform on real-world regression tasks with empirical point cloud data. Can one, for example, use the proposed methodology to study the recorded data presented in the seminal work of Bialek et al. (2012)?

It’d be nice if the authors could comment on the applicability of their method to empirical data.

*References:*
- Statistical mechanics for natural flocks of birds. Bialek et al. (2012)

**Questions:**

1. Neural ODEs are well known to be very difficult to train (see e.g. Dupont et al. (2019), Finlay et al. (2020), Choromanski et al. (2020) or Pal et al. (2021), just to cite a few). Can you comment on how difficult (or easy) was to train your Latent ODE network on the synthetic data you studied? What about the training time?

2. How did you parametrize the decoder network? I don’t find it in the manuscript.

3. Did you consider adding some noise to your observations in point cloud space? As presented, your network should easily be able to handle noisy observations. Similarly, how does the persistent homology representation deal with noise? These questions are of course relevant if one wants to apply your methods to empirical data.

4. How many points from the latent path are used as input to the regressor model? Does the model work with a single point, as e.g. the last point along the latent trajectory? It’d have been nice to understand what information of the latent path is important for the regression task, specially given that completely dispensing from the latent dynamics still gives compelling results (i.e. Table 3).

*References:*
- Augmented neural odes.  Dupont et al. (2019)
- How to train your neural ode: the world of jacobian and kinetic regularization. Finlay et al. (2020)
- Ode to an ode. Choromanski et al. (2020)
- Opening the blackbox: Accelerating neural differential equations by regularizing internal solver heuristics. Pal et al. (2021)

**Limitations:**

Yes, they did address the limitations of their method.

---

> ### Author Rebuttal · Authors · 2024-08-06
>
> For better readability, we restate your comments/questions in *italic*, our response(s) are marked by &#9654;
>
> **Ad contribution:** The reviewer is correct in that we use the latent ODE framework of (Rubanova et al., 2019). However, the latter is only one particular variant (of many; other options are, e.g., latent SDEs or Continuous Recurrent Units (CRUs)) for modeling latent dynamics. Our main contribution is the idea of modeling the point cloud dynamics through the lens of vectorized topological summaries, i.e., capturing topological changes on a *persistence diagram level*, as opposed to capturing changes on the level of individual points in such topological summaries. Even this arguably "simple" approach yields remarkable predictive performance for the parameters of governing equations of collective behavior. Furthermore, our approach avoids the need to track particles over time (and infer velocities) or to rely on computationally expensive summaries of time-dependent persistence.
>
> **Ad empirical data:** Thank you for that comment. Yes, our approach would work on the data from (Bialek et al., 2012). In fact, one motivation for our work was to alleviate the challenge of having to compute/estimate per-particle velocities (this is required in the work of Bialek et al. as they compute correlations from normalized velocities). In many real-world settings, one might not even be able to unambiguously track particles (let alone when the cardinality of the point clouds may vary as well). Instead, our approach only hinges on the position of the particles (and can even naturally deal with point clouds of varying size, as demonstrated with the `volex-10k` experiment).
>
> **Ad training time**: Please see our runtime analysis in the *General Response* section.
>
> *Neural ODEs are well known to be very difficult to train (see e.g. Dupont et al. (2019), Finlay et al. (2020), Choromanski et al. (2020) or Pal et al. (2021), just to cite a few). Can you comment on how difficult (or easy) was to train your Latent ODE network on the synthetic data you studied? What about the training time?*
>
> &#9654; First, we did not experience any difficulty training the full model. However, we did only experiment with the arguably simplest ODE model (an autonomous ODE). We did preliminary experiments also with a non-autonomous variant, but did not observe any noticeable improvements. Nevertheless, our experiments support the conclusion that even the "simplest" choice of latent ODE already suffices to largely outperform existing techniques.
>
> *How did you parametrize the decoder network? I don’t find it in the manuscript.*
>
> &#9654; We apologize for not being clear enough on that point. The decoder (aka reconstruction network, denoted as $Dec_\\gamma$ in Fig. 2 of the manuscript) is a simple 2-layer MLP with ReLU activation that maps latent states (from the latent ODE) to reconstructed persistence diagram vectorizations. We will include this information in the revised manuscript's appendix (and point to the locations in our source code).
>
> *Did you consider adding some noise to your observations in point cloud space? As presented, your network should easily be able to handle noisy observations. Similarly, how does the persistent homology representation deal with noise? These questions are of course relevant if one wants to apply your methods to empirical data.*
>
> &#9654; Currently, only the governing equations of the Vicsek model (see Fig. 3) include noise through the Brownian motion. In general, we point out that due to the stability of the persistence diagrams wrt. perturbations of the input (see the *Stability/Continuity aspects* section of the manuscript), our method is suitable in case of observation noise (unless the noise is excessively large). E.g., on `dorsogna-1k` where points are within $[-1,1]^3$, adding Gaussian noise $\\mathcal{N}(0,\\sigma)$ to all point coordinates and all time points, ($R^2$,SMAPE) drops from (0.846 $\\pm$ 0.011, 0.097 $\\pm$ 0.005) to
> (0.822 $\\pm$ 0.016, 0.098 $\\pm$ 0.002) at $\\sigma=0.01$, and to (0.730 $\\pm$ 0.013, 0.150 $\\pm$ 0.001) at $\\sigma=0.1$.
>
> *How many points from the latent path are used as input to the regressor model? Does the model work with a single point, as e.g. the last point along the latent trajectory? It’d have been nice to understand what information of the latent path is important for the regression task, specially given that completely dispensing from the latent dynamics still gives compelling results (i.e. Table 3).*
>
> &#9654; Again, we apologize for not being precise enough. In fact, we experimented with multiple variants: you are correct that one could use the “last” point along the latent trajectory. Another variant would be summarizing the latent trajectory via a signature approach (although this summary can become quite high-dimensional). The most conceptually elegant approach, from our perspective, was to re-use the encoder architecture (i.e., a duplicate of the mTAN encoder module with its own set of parameters, denoted as $Enc_{\alpha}$ in Fig. 2). However, instead of receiving an unequally-spaced sequence of vectorized persistence diagrams, $Enc_{\alpha}$ receives ALL points along the latent trajectory. In our case, “all” means 100 latent states at equally spaced time points in $[0,1]$, which we obtain by integrating the latent ODE forward in time.
>
> Also, our baseline (w/o dynamics) could be considered quite strong since it uses an mTAN encoder (i.e., an attention-based approach) that directly maps sequences of vectorized persistence diagrams to parameter predictions. While mTANs can attend to relevant information in a sequence, this baseline still consistently falls short of our approach, which explicitly models the dynamics.

---

> ### Comment · Reviewer_bRy6 · 2024-08-11
>
> I thank the authors for their detailed responses. After reading the reviews from other reviewers and your replies to them, I have decided to increase my score.
>
> I believe that, if the updated manuscript addresses all the points raised during this discussion session, it will make a valuable contribution.

---

### Author Rebuttal · Authors · 2024-08-06

# General Response

We like to thank **all** reviewers for their overall positive feedback, their time, and their valuable comments and suggestions!

While we address all issues point by point per reviewer, we first comment on our approach's *computational aspects* and present a detailed runtime analysis as this issue has come up across multiple reviewers.

---
For reference, we refer to the following works in our rebuttal:

**(Rubanova et al., 2019)** Y. Rubanova, R.T.Q. Chen, and D. Duvenaud. Latent ODE for irregularly-sampled time series. In: NeurIPS 2019.

**(Hofer et al., 2019)** C. Hofer, R. Kwitt, and M. Niethammer. Learning representations of persistence barcodes. In: JMLR 20.126 (2019), pp. 1–45.

**(Bialek et al., 2012)** W. Bialek, A. Cavanga, I. Giardina, and A. Walczak. Statistical mechanics for natural flocks of birds. In: PNAS 109.13 (2012), pp. 4786–4791.

**(Carriere et al., 2021)** M. Carriere, F. Chazal, M. Glisse, Y. Ike, H. Kannan and Y. Umeda. Optimizing persistent homology based functions. In: ICML 2021.

**(Oelschläger, 1990)** K. Oelschläger. Large systems of interacting particles and the porous medium equation. In: Journal of Differential Equations, Volume 88, Issue 2.

---

## Runtime analysis (pre-processing)

We re-ran experiments on the `dorsogna-1k` dataset (1k sequences of length 100) and provide a breakdown of runtime below (measured on the same system as described in Appendix C of our manuscript):

First, we point out that computing Vietoris-Rips persistent homology (PH) as well as persistence diagram (PD) vectorization is done as a *pre-processing step* for all available point cloud sequences. *These steps are trivially parallelizable across multiple CPUs/GPUs.*

**Vietoris-Rips PH computation.** The following table contains wall clock time measurements (using Ripser++ on one GPU) per point cloud. We list runtime for computing $H_0$ and $H_1$ features, as well as for computing $H_0, H_1$ and $H_2$. Point clouds are of size 200, as in the manuscript:

|                              | Time per point cloud  | Overall          |
|---                           |---                    |---               |
| PH ($H_0$, $H_1$)            | 0.018 s               | $\approx$ 30 min |
| PH ($H_0$, $H_1$ and $H_2$ ) | 0.330 s               | $\approx$ 6 hrs  |

**PD vectorization.** Vectorization can essentially be broken down into two steps: (i) parameter fitting for the structure elements of (Hofer et al., 2019) and (ii) mapping PDs to vector representations using those structure elements. The following table lists the runtime for both steps on `dorsogna-1k` when vectorizing 0-, 1- and 2-dimensional persistence diagrams:

|                                | Time per diagram  | Overall        |
| ---                            | ---           | ---            |
| (i) Parameter fitting          | n/a           | $\approx$ 27 s |
| (ii) Mapping of PDs to vectors | 0.0052 s      | $\approx$ 52 s |

*PH computation & PD vectorization are both necessary steps for our approach, Crocker Stacks, as well as the PSK approach*.

## Runtime comparison to PRIOR WORK (using topological summaries)

Below, we list the overall **training times** for our approach, Crocker Stacks, and the PSK method. Runtime for pre-processing (see above) is *excluded* from these measurements. Also, PSK and Crocker Stack timings do include hyperparameter optimization, as suggested in the corresponding references. Importantly, this is *not* optional but required to obtain decent regression performance:

|                    | Time         |
| ---                | ---          |
| **Crocker stacks** | 24600 s (6 hrs 50 min) |
| **PSK**            | 646 s  |
| **Ours** (PH-only) | 190 s   |

Notably, PSK kernel computation scales quadratically with the number of sequences $n$, kernel-SVR training takes time somewhere between quadratic and cubic, hence scaling up the number of training sequences $n$ quickly becomes computationally prohibitive for the PSK method, especially in light of the required hyperparameter tuning. Finding suitable hyperparameters is also the main bottleneck for Crocker Stacks (which rely on a linear SVR).

## Runtime comparison to the BASELINE model (w/o dynamics)

We also compare the runtime of our approach to the *baseline* model which does not
explicitely model any dynamics via a latent ODE.

|                                                    | Time   |
| ---                                                | ---    |
| **Ours** (PH only)                                 | 190  s |
| **Ours** (PointNet++ only)                         | 3780 s |
| **Ours** (PH & PointNet++)                         | 4100 s |
|                                                    |        |
| **Baseline (w/o dynamics)** (PH only)              | 50 s   |
| **Baseline (w/o dynamics)** (PointNet++ only)      | 525 s  |
| **Baseline (w/o dynamics)** (PH & PointNet++)      | 600 s  |

We will include such a runtime study (across datasets) in our appendix.

---

### Decision · Program_Chairs · 2024-09-25

**Decision:**

Accept (poster)

**Comment:**

The paper addresses the problem of modeling the dynamic evolution of point clouds based on partial observations at different time steps. This evolution is captured in a latent representation space using an encode-process-decode framework. The latent representation leverages concepts from persistent homology of point clouds and their vectorization. To achieve this, the paper employs an existing vectorization technique that is Lipschitz continuous with respect to the Wasserstein distance. The evolution of the point cloud's vectorized topological representations is modeled using a Neural-ODE-like formulation. This model is then applied to an inverse problem: inferring the parameters of a prior physical model, specifically the governing equations underlying the point cloud observations. The proposed model outperforms recent baselines, including the vision model PointNet, which operates directly in the observation space.
The main originality of the work lies in the combination of vectorized topological representations of point clouds and Neural ODEs to solve an inverse problem related to modeling collective behaviors. While the neural network architecture follows an encode-process-decode framework, which is now standard for modeling dynamical systems (e.g. PDEs), its application in the context of collective dynamics is novel.

All reviewers agree on the originality of the approach and the significance of the contribution. Modeling the evolution of point clouds using topological summaries, which capture global features, is considered a novel and interesting idea. The authors have thoroughly addressed the main concerns raised by the reviewers and added new experimental results that further reinforce the contribution. The paper would benefit from additional technical details and some rewriting to make it more accessible to readers who are not specialists in persistence diagrams and their vectorization. The authors are encouraged to consider these reviewer comments. All reviewers are in favor of acceptance.